# Comparative Analysis of C-Reactive Protein Levels in the Saliva and Serum of Dogs with Various Diseases

**DOI:** 10.3390/ani10061042

**Published:** 2020-06-17

**Authors:** Yoo-Ra Cho, Ye-In Oh, Gun-Ho Song, Young Jun Kim, Kyoung-Won Seo

**Affiliations:** 1Department of Veterinary Internal Medicine, College of Veterinary Medicine, Chungnam National University, Daejeon 34134, Korea; yoora753@naver.com (Y.-R.C.); yeniyen25@gmail.com (Y.-I.O.); songkh@cnu.ac.kr (G.-H.S.); 2Diagnosis and Therapy Lab., Welfare & Medical Division, Electronics and Telecommunications Research Institute, 218 Gajeong-ro, Yuseong-gu, Daejeon 34129, Korea

**Keywords:** C-reactive protein, dogs, saliva

## Abstract

**Simple Summary:**

When collecting blood samples in dogs for evaluating inflammatory conditions, this method is invasive and stressful to dogs. We compared the C-reactive protein (CRP) level in both blood and saliva samples to investigate the non-invasive test method for inflammation. There was a correlation between the blood and saliva CRP levels. This non-invasive test using saliva seems to be a useful method with which to assess inflammatory conditions in dogs.

**Abstract:**

We performed this study to characterize the difference between the inflammatory and non-inflammatory status in diseased dogs by measuring salivary C-reactive protein (CRP) levels. In addition, we assessed whether a correlation exists between CRP levels in saliva and those in serum. CRP levels were measured in 32 client-owned dogs, which were then divided into inflammation and non-inflammation groups based on the serum CRP level. The salivary CRP level was higher in the inflammation group than in the non-inflammation group (*p* < 0.05). Furthermore, there was a positive correlation between the salivary and serum CRP levels (*R* = 0.866, *p* < 0.001). These data suggest that canine salivary CRP measurements can effectively and non-invasively detect an inflammatory state in dogs.

## 1. Introduction

In dogs, C-reactive protein (CRP) is a major acute phase protein produced by non-specific tissue injury [1]. Its resting serum concentration is low; however, it could increase rapidly after exposure to inflammatory stimuli, then decrease after the resolution of inflammation [2]. Due to these characteristics, CRP comprises a sensitive and specific biomarker of systemic inflammation that is useful for the diagnosis and monitoring of various inflammatory diseases [3,4].

CRP levels are typically measured using serum collected by venipuncture, a painful and stress-inducing procedure that requires the involvement of professional and skilled clinicians or technicians, as well as laboratory equipment. Saliva collection is non-invasive, low-stress, and pain-free; therefore, it is an attractive alternative method for the evaluation of individual immune activity via CRP levels [5]. Saliva is an easily available biological fluid that contains many local and systemic factors that enter the oral cavity [6]. In humans, saliva samples have been successfully used for the detection of several biomarkers, such as cortisol and alpha-amylase [7,8]. In addition, saliva samples have been used in several studies of cortisol, catecholamine, and phenobarbital in veterinary medicine [9,10,11]. A previous study measured the CRP level from saliva and serum in dogs [12]. In this study, salivary CRP and serum CRP show a positive correlation, and it is considered that there is a significant difference between healthy and inflamed dogs. However, it has not been studied in diseased dogs with no inflammation. 

The objectives of this study were to examine the differences in the salivary CRP levels between diseased dogs classified into inflammation or non-inflammation groups, without the use of chemical stimulants. Moreover, we assessed whether a correlation exists between CRP levels in serum and those in saliva.

## 2. Materials and Methods

### 2.1. Animals and Sample Collection

Thirty-two client-owned dogs with various diseases were presented to the Veterinary Medical Teaching Hospital at Chungnam National University during the period from May 2017 to September 2017 and were included in our study. Clients were informed of this study and agreed to participate. All dogs underwent a complete physical examination, blood count (Advia2120; Siemens Healthcare Diagnostics, Deerfield, IL, USA), CRP measurement, and serum biochemistry profile (Mindray BS-300; Mindray Bio-Medical Electronics Co., Ltd., Shenzhen, China). Dogs were classified into inflammation or non-inflammation groups according to the laboratory results: dogs with serum CRP values higher than the reference range of 0–2 mg/dL were included in the inflammation group, whereas dogs with lower serum CRP values were included in the non-inflammation group. Both serum and saliva specimens were collected from each dog at a single time point. Before collection of saliva samples, dogs were fasted for 6–12 hours. No acid stimulants were used to increase salivary secretion. Dogs that had apparent periodontal disease or bleeding gums in the physical examination were excluded from the study.

Blood samples were obtained from each dog by venipuncture and collected in tubes containing a coagulation activator. Blood was allowed to clot at room temperature and centrifuged at 3000× *g* at room temperature for 10 min. Serum samples were stored at −70 °C until the biochemical analysis. After taking a blood sample, saliva was obtained using a cotton pad and a 1.5-mL tube (Greiner Bio-One, Kremsmünster, Austria). The cotton pad was placed in the oral cavity in contact with the buccal mucosa, or the dogs were allowed to chew the cotton pad for 1–2 min. The pad was then placed in the tube and centrifuged for 15 min at 3000× *g*. Immediately after centrifugation, salivary samples were stored in Eppendorf tubes and frozen at −70 °C until the biochemical analysis. All saliva and serum samples were frozen in aliquots, and only vials needed for each assay were thawed to prevent any potential variation as a result of repetitive freeze-thaw cycles. The sample collection protocol was approved by the Institutional Animal Care and Use Committee at Chungnam National University (approval number CNU-00950).

### 2.2. Measurement of CRP Concentration in Both Saliva and Plasma

CRP levels in both the serum and saliva samples were measured using a commercial canine CRP ELISA kit (Abcam, Cambridge, MA, USA). Dilutions of the serum and saliva samples used in analyses were 1:1000, 1:10, and undiluted. All thawed salivary samples were centrifuged at 1500× *g* for 15 min at 4 °C to remove cellular debris and minimize the turbidity of the saliva, which could negatively impact the accuracy of analysis [13]. Supernatants were transferred into fresh Eppendorf tubes and appropriately labeled. A total of 64 samples, consisting of 32 serum and 32 saliva samples, were assayed in duplicate. For ELISA analysis, all samples were thawed and mixed thoroughly; ELISA was then performed in accordance with the manufacturer’s instructions. Absorbance was detected at a wavelength of 450 nm on a microplate reader (Biotek Instruments Inc., Winooski, VT, USA).

### 2.3. Statistical Analysis

Statistical analyses were performed using IBM SPSS software (version 24, IBM Corp., Ehningen, Germany). CRP levels were expressed as both mean and standard deviation by Kolmogorov–Smirnov test. Pearson’s product-moment correlation and simple regression analysis was used to compare CRP levels in saliva and serum. Differences in salivary CRP were analyzed using unpaired t-tests and F-test was used to compare variances. Differences were considered statistically significant when *p* < 0.001.

## 3. Results

The 32 client-owned diseased dogs consisted of 15 different breeds, including Maltese (*n* = 9), Shih Tzu (*n* = 4), Schnauzer (*n* = 3), Yorkshire Terrier (*n* = 2), Welsh Corgi (*n* = 2), Pomeranian (*n* = 2), Beagle (*n* = 2), Sapsaree (*n* = 1), Poodle (*n* = 1), German Shepherd (*n* = 1), French Bulldog (*n* = 1), Cocker Spaniel (*n* = 1), Chihuahua (*n* = 1), Cane Corso (*n* = 1), and Australian Shepherd (*n* = 1). Based on a general health screening and laboratory evaluation, the dogs were diagnosed with various diseases. Animals were divided into two groups (inflammation and non-inflammation) according to CRP seric levels, regardless of the conditions present. Cardiologic disease was present in a large proportion of the non-inflammation group. Well-managed or congenital cardiological disease and acute pancreatitis (recovered) were included in the non-inflammation group. Post-operative inflammation within four days and acute pancreatitis at the time of diagnosis were the most common conditions in the inflammation group. The clinical characteristics of all dogs are shown in Table 1 and Table 2. Salivary CRP levels were significantly higher in the inflammation group than in the non-inflammation group. Furthermore, there was a positive correlation between the serum and salivary CRP levels in all dogs (R = 0.866, *p* < 0.001; Figure 1).

## 4. Discussion

The results of this study indicate that salivary CRP is an accurate method for identifying inflammation in diseased dogs. Saliva can be collected without the use of invasive methods, and is advantageous for pediatric patients or those with a specific clinical pathologic state (e.g., anemia or hemostatic disease), physical sensitivity (e.g., to needles), and geographic handicaps (e.g., those who reside far from the hospital) [6,12].

In this study, salivary CRP was able to serve as an indicator of inflammatory status and showed a moderate-to-strong association (*R* = 0.866 and *R*^2^ = 0.749) with the serum CRP level. The quantification of salivary CRP and its correlation with the serum CRP has been assessed in human [5], porcine [14], and canine [12] studies. Although this correlation in dogs (*R*^2^ = 0.69) was higher than that in humans (*R*^2^ = 0.49) or pigs (*R*^2^ = 0.52), a moderate association was detected in all groups.

We observed a difference of the salivary and serum CRP levels which do not meet the regression equation. The low ratio of the saliva CRP level to the serum CRP level could reduce the precision of salivary CRP measures, particularly at low serum CRP concentrations in a non-inflammation state [5]. Although the mechanisms by which CRP is transported from serum to saliva remain unclear [5], CRP could potentially enter in the serum in its high-molecular weight form, and its two glycosylated subunits could increase its lipid-insolubility [15,16], preventing CRP from entering saliva. For large molecules such as proteins and charged steroids, the primary route of entry into the oral cavity is through plasma exudates of systemic origin from gingival crevicular fluid (GCF) [17], as well as from minor abrasions in the mouth, instead of diffusion or ultrafiltration [9,18]. Whole saliva used in this study comprised oral fluids from the major and minor salivary gland, as well as fluids of non-salivary origin, including GCF, serum transudate from the mucosa and sites of inflammation, epithelial and immune cells, food debris, and many microbes [18,19]. Poor oral health in diseased dogs might have caused minor bleeding or GCF overflow from micro-injuries, which could have increased salivary CRP in the absence of elevated serum CRP. Furthermore, steroid hormones are known to be metabolized by oral bacteria and epithelial cells in the salivary gland during transcellular movement [20]. Although this mechanism is not known in salivary CRP, this modification and subsequent metabolism could considerably modify or reduce detection in saliva.

Our investigation has different experimental settings and collection methods compared to previously published studies. The composition of whole saliva can be rapidly altered by flow rate, the degree of stimulation in various glands, and the time of day [21]. Our protocol controlled for animal factors by fasting, acid stimulant exclusion, and collection time, as well as by the exclusion of dogs with periodontitis and other oral diseases. In addition, we collected samples before treatment, which may have increased the association with inflammation. A previous study in humans reported that mechanical stimulation, but not acid stimulation, is the most viable option for acquisition of saliva without changes in CRP levels [21]. Based on these results, we used mechanical stimulation for saliva collection in our study. Further studies in veterinary medicine are needed to confirm the optimal collection protocol.

There were some limitations in the present study. Firstly, we included a relatively small number of dogs. Most diseased dogs who were presented to the clinic showed signs of dehydration and dry mucosa in the oral cavity; therefore, we were unable to obtain a sufficient volume of saliva for analysis. Secondly, our study used sample pretreatment (centrifugation) to remove cellular debris. Processing methods can adversely affect the concentration of CRP in humans [21]; therefore, it is necessary to verify whether the same adverse effect is present in veterinary medicine samples.

## 5. Conclusions

Taken together, the results of this study suggest that salivary CRP measurements in dogs can provide a straightforward, adaptable, and non-invasive method for the detection of inflammation. This method could be used as a tool to monitor health and assess inflammation in diseased animals. To use saliva for the assessment of inflammation in clinical practice, further studies should focus on creating reference ranges in different species, as well as the stability of saliva samples in various storage conditions.

## Figures and Tables

**Figure 1 animals-10-01042-f001:**
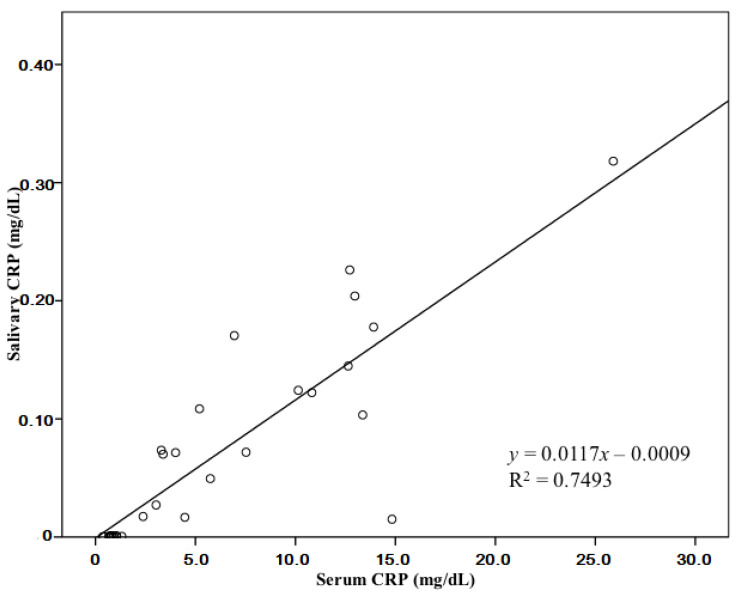
Correlation between C-reactive protein (CRP) levels in serum and saliva.

**Table 1 animals-10-01042-t001:** Measured C-reactive protein (CRP) levels in the serum and saliva samples and the clinical conditions of the inflammation group (*n* = 19) and non-inflammation group (*n* = 13).

**Non-Inflammation Group**			
**No.**	**Serum CRP Levels** **(mg/dl)**	**Salivary CRP Levels** **(mg/dl)**	**Clinical Condition**
1	0.6938	0.00060	Peripheral vestibular syndrome
2	1.3186	0.00054	Acute pancreatitis (Being recovered)
3	0.6841	0.00048	Urolithiasis (No cystitis)
4	0.3895	0.00038	Mast cell tumorectomy (2 years ago)
5	0.3772	0.00037	Mitral valve insufficiency (Being treated)
6	0.9053	0.00073	Mitral valve insufficiency (Being treated)
7	0.7934	0.00069	Hyperthyroidism
8	1.0628	0.00085	Tricuspid valve insufficiency (Being treated)
9	0.9215	0.00075	Lymphoma (Being treated)
10	1.0376	0.00094	Patent ductus arteriosus
11	0.7037	0.00070	Patent ductus arteriosus
12	0.8106	0.00076	Atopic dermatitis (Being treated with oclacitinib)
13	0.7889	0.00010	Narcolepsy
**Inflammation Group**			
**No.**	**Serum CRP Levels** **(mg/dl)**	**Salivary CRP levels** **(mg/dl)**	**Clinical Condition**
1	10.14104	0.12419	Orthopedics
2	14.83298	0.01509	Thyroid tumor
3	12.71612	0.22608	Gastrointestinal stromal tumor, peritonitis
4	6.94354	0.17053	Thyroid tumor
5	13.36547	0.10334	Acute pancreatitis, anaplasmosis
6	10.82076	0.12226	Cystotomy, nephrotomy
7	12.63759	0.14473	Orthopedics
8	13.91174	0.17774	Orthopedics
9	12.97129	0.20399	Tumorectomy
10	3.03367	0.02713	Tumorectomy
11	5.74407	0.04943	Trauma
12	5.20193	0.10855	Acute pancreatitis
13	3.29112	0.07348	Cardiogenic pulmonary edema, mitral valve insufficiency
14	4.46760	0.01669	Pyometra, mammary gland tumor, acute pancreatitis
15	2.38347	0.01738	Urolithiasis, cystitis
16	4.00526	0.07147	Acute colitis
17	25.89826	0.31818	Pneumonia, Chronic kidney disease
18	7.53124	0.07178	Orthopedics

**Table 2 animals-10-01042-t002:** Comparison of the signalment and median CRP levels of the inflammation group and non-inflammation group.

	Inflammation Group (*n* = 19)	Non-Inflammation Group (*n* = 13)
Signalment		
Age ^1^	9 (2–18)	11 (1–16)
Sex ^2^	3M, 7CM, 6F, 3SF	1M, 10CM, 1F, 1SF
CRP levels (mg/dL)		
Serum ^3^	9.11994 ± 0.11117 *	0.80669 ± 0.00061
Saliva ^3^	5.75988 ± 0.07887	0.24827 ± 0.00022

^1^ Age is represented in years as mean (range). ^2^ Sex is represented as the number of dogs in each subcategory of sex. ^3^ Serum and saliva are represented as mean ± standard deviation. * Significant statistical difference between the inflammation group and non-inflammation group indicates *p* < 0.0001. M, male; CM, castrated male; F, female; SF, spayed female; CRP, C-reactive protein.

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
