# Peer review of "Comparative Analysis of C-Reactive Protein Levels in the Saliva and Serum of Dogs with Various Diseases"

_animals, 2020, doi:10.3390/ani10061042_

Round 1
Reviewer 1 Report
Authors have adressed my concerns
Author Response
Thank you very much.
Reviewer 2 Report
The revision version is complete.
Author Response
Thank you very much.
Reviewer 3 Report
Methods
The authors chose not to characterize the sample clearly. A table with the conditions/ diagnosis of the animals included in both groups should be included.
There is evidence that a number of cardiological and neurological diseases have inflammatory aspects so it is relevant to know which diseases affect these animals.
For example, degenerative valvular disease, the most common acquired heart condition in dogs, is known to be characterized by increased IL-6 and IL-1 expression (Oyama et al, 2006; Markby et al, 2017); and IL8 seric levels (Mavropoulou et al 2016). CRP seric levels have also been described as elevated in dogs with degenerative valvular disease (Rush et al, 2008); on the other hand, as the authors state these "were the most common conditions", it is obscure which conditions were left out.
Age, gender and reproductive state is not a thorough "clinical characterization".
A complete and clear sample characterization is essential.
The authors should state which test was used to assess data normal distribution.
The phrase "Clients were obtained information of this study and agreed participation." is not correct. It could be changed into "Clients were informed on this study and agreed to participate" or "Animal owners' informed consent was obtained previously to enrollment in the present study".
Round 2
Reviewer 3 Report
This is an interesting study if one considers the potential of saliva CRP in comparison to seric levels, in animals with various diseases, but how it is presented must be considered carefuly.
The authors' effort to better characterized the sample is appreciated. However, it is difficult to support the inclusion of acute pancreatitis, atopic dermatitis cases in the non-inflammation group, just to mention the two more obvious diseases.
The authors mention these animals were being treated but influence of administered drugs on inflammatory response in some of these patients is not considered (have NSAIDs been given to some of these patients (eample, for pain control in acute pancreatitis? How was atopic dermatitis being treated? Were steroids or JAK1 inhibitors used? Was lokivetmab used?). Furthermore, although I understand bacterial infection may be excluded in urolythiasis, I see with difficulty that inflammation of the urinary tract would be completely absent, unless, again, medications were given that downplayed the inflammatory response.
It would more correct to say animals were divided into groups accordingly to CRP seric levels, regardless of the conditions present.
Author Response
Please see the attachment.

This manuscript is a resubmission of an earlier submission. The following is a list of the peer review reports and author responses from that submission.
Round 1
Reviewer 1 Report
In humans, C-reactive protein (CRP) assay in the saliva has been recently proposed as a non invasive method for determining inflammation status in a wide variety of disease conditions.
Its application also for dog is welcome. This study only shows that salivary CRP is significantly increased in dogs with inflammation and that a very good correlation exists between serum and salivary CRP.
Authors, however did not asses the sensitivity, specificity and likelihood ratio of the assay. This is the major limit of this study.
Statistics needs to be implemented. Is the data distributed normally? If they are, then the tests applied are fine but it would be better to change the description of the data in the table, and instead of median and IQR put average and SD; if they are not, then the table is right but Spearmann's test must be applied (not parametric and more restrictive than parametric ).
Reviewer 2 Report
After reviewing the manuscript entitled with " Comparative analysis of C-reactive protein levels in the saliva and serum of dogs with various diseases". The study characterizes the dog inflammatory and non-inflammatory status by measuring salivary C-reactive protein levels assessing whether exists a correlation between saliva and serum CRP levels. I have concluded that the manuscript is well written and is in a relevant field. However, there is a point in the used method that is not appropriate; the reduced number of animals included in the trial. To consider a higher statistical power of a hypothesis, the inflammation group (n=19) and non-inflammation group (n=13) should be increased.Reviewer 3 Report
C-Reactive Protein (CRP) is a non-specific inflammation biomarker. It has been measured in dog saliva, both in healthy and diseased animals (Parra et al., 2005). In this manuscript, the authors explore the possibility of differentiating non-inflammatory and inflammatory conditions in dogs by CRP measurement in saliva.
However, there are a few points that should be better clarified.
In introduction, I believe the work from Parra et al, 2005 should be duly referred. Differences between both studies should be highlighted in this section so that the relevance of the present study could be clarified.
In Materials and Methods it would be necessary to clarify:
- Whether animal owners gave their informed consent for inclusion in this study;
- It would be very important to have a thorough sample characterization, namely by a table that listed the animals diagnosed conditions in both groups and fasting times since 6 to 12 hours fasting is a time interval of some importance and should be considered in sample characterization.
- If saliva was collected before the blood sample or after blood collection or whether no specific order of procedures was observed
- A healthy animals control group is desirable.
- Why opt for just 1:1000 and 1:10 dilution, without considering a 1:100 seriated dilution point? Was there some form of exploratory previous study done that explained why a mid-point dilution was not included?
- Other authors referred difficulties obtaining canine saliva from cotton wool pads due to high absorption. Was this a difficulty the present study? How much saliva per animal was it possible to recover, in average?
- Concerning the statistical analysis, the independent samples t-test depends on two sets of equally distributed samples. The groups are unequally sized. The authors fail to mention which tests were performed to assess variance homogeneity.
- The authors mention that a difference of salivary and serum CRP levels did not meet the regression equation but still chose to present results through a linear model, leading to conclusions that may be not entirely supported by data. Did the authors considerer a polynomial regression model?